# Seasonal *Paspalum vaginatum* Physiological Characteristics Change with Agricultural Byproduct Biochar in Sandy Potting Soil

**DOI:** 10.3390/biology11040560

**Published:** 2022-04-07

**Authors:** Dounia Fetjah, Zaina Idardare, Bouchaib Ihssane, Lalla Fatima Zohra Ainlhout, Laila Bouqbis

**Affiliations:** 1Laboratory of Biotechnology, Materials, and Environment, Faculty of Sciences, Ibn Zohr University, Agadir 80000, Morocco; 2Laboratory of Biotechnology, Materials, and Environment, Higher Institute of Maritime Fisheries, Agadir 80000, Morocco; idardarezaina@yahoo.fr; 3Laboratory of Applied Organic Chemistry, Faculty of Sciences and Techniques, Sidi Mohamed Ben Abdellah University, Fez 80000, Morocco; bouchaib.ihssane@usmba.ac.ma; 4Laboratory of Biotechnology, Materials, and Environment, Faculty of Applied Sciences, Ibn Zohr University, Ait Melloul 86150, Morocco; ainhfz06@gmail.com

**Keywords:** seashore paspalum, seasonal physiological parameters, MFA, FAMD, peanut hull biochar, walnut shells biochar, almond shells biochar

## Abstract

**Simple Summary:**

Understanding the changes in physiological parameters of *Paspalum vaginatum* is important for scientists and golf managers to maintain adequate irrigation and nutrient levels for this plant under harsh conditions. The south of Morocco is an area of the country well known for its luxury golf courses, even though the region has a dry climate due to low rainfall. Irrigation of its golf courses is a major issue in that region, particularly during the summer. Our goal was to see if the three biochars derived from agricultural wastes could improve the ecophysiological responses of *P. vaginatum* under adequate water supply and moderate drought stress. The effect of peanut hull biochar on improving the physiological and stomatal traits of seashore paspalum under two water shortages was clearer in the summer season compared to the winter and spring seasons in this study. By releasing carbon and organic matter, the addition of 6% peanut hull biochar to soil improved grass tolerance to soil-moisture stress physiology in the summer season. In fact, due to its ability to retain water, the application of peanut hull biochar to soil could potentially save 47.5% on turfgrass in hot, dry climates.

**Abstract:**

A plastic pot open-air trial was conducted with the *Paspalum vaginatum* (seashore paspalum) using different rates of biochar or compost addition to sandy loam soil and two water treatments (60% and 20% of the water-holding capacity of the control) during three seasons (winter, spring, and summer). Paspalum growth, physiological characteristics, and physicochemical properties of soil were investigated. The effect of biochar on soil properties was assessed using factor analysis of mixed data (FAMD). Additionally, multiple factorial designs (MFA) were used to examine the impact of three biochars on physiological functions. Peanut hull biochar application increased soil fertility and chlorophyll concentration of paspalum leaves significantly compared to the other biochars. Physiological characteristics were significantly improved with peanut hull biochar under summer compared to winter and spring due to the accumulation of nutrients in the soil by the decomposition of biochar. The application rate of the three biochars reduced the water requirements of paspalum. The best result was obtained by incorporating 6% peanut hull biochar into the soil, which resulted in better soil quality and healthy grass in dryland conditions while using 47.5% less water. These findings can be suitable for golf managers and can serve as a solution for dry zones.

## 1. Introduction

The amount of organic matter in soil is an important indicator of its quality [1]. Because of a lack of organic matter, arid soils have generally poor quality [2]. Morocco is one of the countries that has suffered from years of drought, soil degradation, and climate change. The water resources of some Moroccan arable lands are depleting due to a lack of rain, and the quality of its soils is deteriorating [3]. Significant effort is required to add organic materials to these soils. Recent techniques have been developed to enhance soil organic matter. Indeed, one such novel technique is the conversion of locally available waste biomass materials into biochar for soil application. Pyrolysis produces not only gases and bio-oil but also biochar that can be used as a soil amendment [4]. Biochar is a tough organic carbon (C) material made by pyrolyzing biomass in low oxygen environments [5]. Because biochar generates very stable organic carbon, incorporating it into the soil has the potential to improve soil quality while also sequestering carbon [6]. Biochar application to the soil has shown several positive effects on soil quality. The positive benefits of biochar amendment on soils include: increasing soil capacity to accumulate plant nutrients, enhancing soil cation exchange capacity [7], developing plant-available water retention [8], and expanding crop production [9]. Agricultural residues are frequently the most readily available biomass feedstock in arid agricultural communities, and freshwater is frequently the most pressing need in these regions. Morocco is an example of an agricultural country with a diverse crop production each year and a high product quality ranking [3]. 

Agricultural residues such as walnut shells, almond shells, and peanut hulls are rich, common sources that can be used as biochar. Furthermore, the country is well known for its high-end golf resorts, which draw visitors from all over the world. Indeed, the Kingdom’s south is currently experiencing the highest levels of water stress and dry conditions. As a consequence, the ability to manage golf courses in arid and semiarid zones may present a serious environmental problem, given the enormous volume of water necessary for irrigating seashore areas. The addition of charcoal to new golf greens for improving porosity and overall properties was first suggested more than 93 years ago [10]. Several recent studies have investigated the use of biochar (agricultural charcoal) as a golf green amendment [11,12]. The implications of walnut shells biochar, almond shells biochar, peanut hull biochar, and compost mixtures on *P. vaginatum* were tested in a pot trial using different amounts of these natural amendments, which were added to sandy loam soil, and water supply treatment options (60% and 20% of the water-holding capacity of the control).

The current research experiment aimed to compare the impact of biochar type on physiological signs and stomatal tendencies using only a variety of qualitative and quantitative methods. Multiple factorial analysis (MFA) was used to investigate the function of three-biochar use on seashore paspalum proliferation using data derived from biological and photosynthetic outcomes collected during three episodes (winter, spring, and summer). The pH, electrical conductivity, and color information of *P. vaginatum* leaf tissue were probed using factor analysis of mixed data (FAMD). The objectives of the current study were to:(i)Evaluate the effect of biochar type from three different local feedstocks (almond shells, walnut shells, and peanut hull) on soil fertility.(ii)Evaluate the role of biochar addition into sandy loam soil and on *P. vaginatum* physiological characteristics in dry climates.

## 2. Materials and Methods

### 2.1. X-ray Diffraction and FTIR

The crystal structure of peanut hull biochar, walnut shells biochar, and almond shells biochar were determined using an X-ray diffractometer [13]. Furthermore, the surface chemistry groups of the three-carbon materials studied were recognized by utilizing Fourier transform infrared (FTIR) spectroscopy [13]. The FTIR spectra of the three biochars studied are depicted in Appendix A). The peak at 3855 cm^−1^ corresponded to the motions of a double bond of C and N atoms inside the benzene ring [14]. OH, and alcohols appeared near 3500 and 3000 cm^−1^ (3495 cm^−1^), C=O or C=C at 1625 cm^−1^ and C–O vibrations at 1118 cm^−1^ deduced from the cellulose portion of biomass residues. In agreement with our findings, Kazemipour et al. [15] realized that the peak at 2358 cm^−1^ indicated the C=N=S or C=N=S groups. Aromatic C–H vibrations affiliated with lignin also were spotted in bending vibrations (867 cm^−1^). The peak at 1832 cm^−1^ is assigned to C=O receiving stretched adsorptions, while the band at 1625 cm^−1^ matches C=C stretching adsorptions. The skeletal C=C vibrations in aromatic rings generate a peak around 1405 cm^−1^ [16]. The 3855 cm^−1^ band pertains to phenol. The region between 700 and 900 cm^−1^ on the same spectra includes various bands linked to aromatic out-of-plane C–H bending with varying degrees of substitution [16]. The final peak, 659 cm^−1^, is exacerbated by O–H out-of-plane trying to bend vibrations [15]. Furthermore, the results of the FTIR technique can be used as an efficient method for managing biochar performance in site conditions, together with its effect on seed germination [14]. Indeed, the FTIR spectra of three agricultural waste biochars were used to deduce functional groups (Appendix A). It revealed the presence of the following groups, with aromatic groups monopolizing the FTIR spectra, particularly when compared to other functions [14]. Because of the predominance of aromatic carbon groups, the three biochars produced from peanut hull, almond shell, and walnut shell biomasses, respectively, may be resistant and have the potential for carbon sequestration. This FTIR result could explain why the three biochars had a positive effect on *P. vaginatum*, even when subjected to underwater stress [13,17].

An X-ray diffraction pattern of activated biochar made from nutshell biochars is shown in Appendix A). The distinctive sharp peaks of the three biochars constitute their crystalline nature and changes in the structure of the biomass prompted by pyrolysis. Between 28° and 40°, the peak XRD pattern for nutshell biochar symbolized silicate minerals such as quartz (SiO_2_), hematite (Fe_2_O_3_), and goethite (Fe_2_O_3_·H_2_O) [18]. The mineral composition is revealed by a large hump with a centroid between 28 and 30 degrees, which confirms the formation of a predominantly crystalline material in WS-B. Scawatite Ca_14.00_Si_12.00_C_2.00_O_46.00_H_8.00_ and carbonate-hydroxylapatite Ca_10.00_P_6.00_O_26.14_H_2.60_C_0.02_ showed the highest activity in the walnut shells biochar. The presence of scawatite was clear in the peak in which the hkl was equal to 240, 003, and 372, corresponding to 2θ equal to 29.564, 45.474, and 66.176°, respectively. Concerning the carbonate-hydroxylapatite phase, it corresponds to the peaks that had 121, 114, 125, and 351 in hkl.

Additionally, the main phase which predominates in the XRD of the peanut hull biochar was Ca_2.00_O_12.00_C_24.00_H_16.00_N_4.00_ at 2θ = 25.996, 28.306, 29.534, 38,952, and 58.852°; for the second phase of peanut hull biochar, the phase was monohydrocalcite Ca_9.00_C_9.00_O_36.00_H_18.00_ at 2θ = 98.708°. For the almond shells biochar, the main phase was from fukalite dimorph at 2θ = 29.454.

As previously stated, the number of peak values increases as sample temperature rises [19].

The peanut hull biochar has a crystalline character, as seen in the XRD graphs (Appendix A). The calcite phase was found in nearly every phase of the three biochars. Due to the obvious crystalline nature of the three agricultural waste biochars, the calcite form assisted the *P. vaginatum* in establishing its growth responses even in harsh conditions (20% WHC), and this was because the calcite form is more soluble by plants. Besides, the calcite formed contributed to the alkalinity of the three biochars studied, as evidenced by their high pH values [13,20].

### 2.2. Biochar Preparation

Biochars were produced from the three biomass feedstocks using a pyrolytic stove, a slow pyrolysis system. Almond shells, walnut shells, peanut hulls were used and converted into biochar. These agricultural byproducts are readily available in Morocco. The system consists of a cylindrical-shaped oven manufactured in Morocco from locally available materials, made of zinc alloy sheet, based on a design provided by Dr. Claudia Kammann (Institute of Plant Ecology, University of Giessen, Giessen, Germany), with a height of 40 cm. The programmable oven consists of different parts: a combustion chamber with a diameter of 18 cm, an outer chamber with a diameter of 28 cm, and a lid with a ventilation tube. Inside the combustion chamber, combustible materials were used for weight reduction purposes. Two circular stainless-steel plates with large holes were held in place with screws; a 40-mesh stainless-steel wire mesh was placed between the plates on the biomass side to contain biomass particles while allowing gas flow. A portable data logger (WINTACT WT900 Infrared Non-Contact Laser Thermometer, USA) was intended to record the biomass temperature every 5 min. The peanut shell biomass began to burn after 30 min at a slow pyrolysis temperature of around 200 °C. In contrast, the almond and walnut biomass began to burn after 40 min at a slow pyrolysis temperature between 200 and 400 °C. The biochar pH for peanut hull, walnut shells, and almond shells was 9.2, 9.41, and 8.91, respectively [3]. The flame turned blue while emitting smoke, indicating that complete combustion of the fuel was achieved. After 2 h, the biomass was completely burned. The resulting peanut shell biochar (33% of the biomass) was then ground and sieved into 2 mm particles to obtain a more homogeneous substrate. The total organic matter content (MOt) of the peanut shell, walnut shell, and almond shell biochar was 92.29, 41.47, and 38.23%, respectively. The 53.53% organic carbon content was very high for peanut hull biochar. The organic carbon contents of the walnut hull and almond hull biochar were 24.05% and 22.18%, respectively (Walkley–Black method). The soil was physicochemically analyzed at the beginning and end of the planting process [21]. The compost was purchased from Agadir.

### 2.3. Experimental Design

The soil for the pot experiment was collected from the south of Morocco, specifically Taroudant, at a deep of 0–20 cm. After air-drying and crushing, it was sieved to obtain 2 mm particles. The soil was then analyzed. It contained 0.007 percent total nitrogen Nt; 0.201‰ P_2_O_5_; 2.5 percent total organic matter MOt; 1.45 percent total organic carbon COt; 0.357‰ K_2_O; 0.192‰ Na_2_O; 0.933‰ CaO; and 0.391‰ MgO. The iron, manganese, copper, and zinc contents were 0.4, 7.4, 0.9, and 2.6 ppm, respectively. The textural class of the soil used for the present experimentation was sandy loam with 35.44% silt, 1.20% clay, and 63.36% sand.

Different preparations were made by soil mixed with organic amendment (biochar and compost): 3C0B (which means 97 percent dry soil mixed with 3 percent compost and 0 percent biochar; 3B0C(97 percent dry soil mixed with 3 percent biochar and 0 percent compost); 3B3C (94 percent dry soil mixed with 3 percent biochar and 3 percent compost); 3B6C (91 percent dry soil, 3 percent biochar, and 6 percent compost); 6B0C (94 percent dry soil, 6 percent biochar, and 0 percent compost); 6C0B (94 percent dry soil, 6 percent compost, and 0 percent biochar); and 6B6C (88 percent dry soil, 6 percent biochar, and 6 percent compost). Additionally, a control (soil without added organic amendment) and a test treatment were prepared by using soil and mineral fertilizer (10–30–10 NPK: 10% N, 30% P_2_O_5_, and 10% K_2_O). Only on the test treatment was fertilizer added.

For the research experiment, an open-air table trial was performed using four replicate pots (four pots for each treatment) under a completely randomized design (CRD). Each experimental unit consisted of a single pot (with a height of 13 cm and a diameter of 12 cm). Each pot contained 2 kg of dry sandy loam soil with the proper weight of amendment thoroughly mixed in. Each of the pots was irrigated with tap water depending on the two water shortages, 20 and 60% water-holding capacity (WHC). WHC was measured for all mixtures: soil–biochar, soil–biochar–compost, and soil-alone treatment [3].

As an experimental plant, we choose grass (*Paspalum vaginatum* Swartz.) mainly because it is relatively undemanding concerning soil quality and climatic conditions. *P. vaginatum* could be grown in the pots, which allowed us to investigate germination and plant survival in golf courses for the case of poor plant growth. Sowing was done using the plastic trays (33.5 cm × 50 cm) to provide nutrients. Then, we transferred the seedlings into a plastic tube. Seashore paspalum was grown differently depending on the soil and organic amendment mix. Seashore paspalum needed 137.04 Mg·ha^−1^ of organic amendments equaling 6% biochar. Seashore paspalum needed 68.52 Mg·ha^−1^ of biochar for 3% biochar. Each pot was provided with 0.25 L of water per day, as required, depending on the two months before applying the two water conditions.

Three experiments were performed depending on necessary soil amendments (seashore paspalum amended with peanut hull biochar and compost mixtures; paspalum amended with almond shells biochar and compost mixtures; and paspalum amended with walnut shells biochar and compost mixtures). Each experiment pot was placed on a table. Indeed, three tables were arranged in a 2-by-2 configuration, with arrangements assigned to the two diagonals at random. In a 10-by-2 grid with ten rows and two columns, 80 pots were assigned to each table. All pots on tables with fixed-position arrangements were randomly assigned to positions on a table once and they were not moved (except for weighing and watering, after which they were replaced to their fixed positions). The pots were randomly rotated in case of cloudy or rainy days to a different position within the block for the duration of the trial. Then, pots were returned to the same position after the inclement days. The experiment was done in Agadir, a city located in the south of Morocco; this city has a dry climate for the entire year. For optimal growth, we measured temperature and relative humidity for each season. For the winter season, which began on 21 December 2019 and continued until 18 March 2020, the maximum temperature was 37.9 °C, the minimum temperature recorded was 5.7 °C, and the average winter season temperature was 17.3 °C. During the spring season, which began on 20 March 2020 and continued until 21 June 2020, the maximum temperature was 52.3 °C, the lowest temperature recorded was 9.4 °C, and the average spring season temperature was 23 °C. For the summer season, which began on 21 June 2020 and continued until 23 September 2020, the maximum temperature was 60 °C, the lowest temperature was 16.8 °C, and the average summer season temperature was 26.1 °C. Concerning the relative humidity for each season, the average relative humidity was 70, 64.2, and 68.3% for winter, spring, and summer, respectively. For further information, the Appendix A provide all measurements collected during the year, recorded by a Data Logger UNI-T UT330A (Appendix A). For solar radiation, during 95 days, the average was 82.04 W/m^2^ from September to December 2019. In 2020, the average solar radiation was 231.07 W/m^2^. These measurements were recorded by the meteorological station at Agadir, Morocco.

Several measurements were taken during this experiment. Four factors (seasons, type of treatment, type of biochar, and irrigation level) were chosen to evaluate their effect on seasonal adaptations of seashore paspalum. We examined a set of measurements of several physiological parameters of paspalum leaves cultivated in the treatments amended with the three biochars (photosynthesis parameters: A = net photosynthetic rate (mmol m^−2^ s^−1^), E = transpiration rate (mmol m^−2^s^−1^), gs = stomatal conductance (mol m^−2^s^−1^), WUE = water-use efficiency (mmol m^−2^s^−1^/mmol m^−2^s^−1^); stomatal traits: LS = stomata length (mm), WS = stomata width (mm), Lo = ostiole length (mm), Wo = ostiole width (mm); anatomical parameters: CPT = chlorophyll parenchyma thickness (mm), TPT = total parenchyma thickness (mm)). FWBA (fresh weight of aerial biomass (g)), DWBA (dry weight of aerial biomass (g)), and relative water content RWC (%) were measured at 6 a.m. and 12 a.m. during three seasons with two irrigation water conditions (20 and 60% WHC). Additionally, the pigment (chla = chlorophyll a (mg·g ^−1^·FW), chlb = chlorophyll b (mg·g ^−1^·FW), chla + b = total chlorophyll (mg·g ^−1^·FW), and car = carotenoids: (mg·g ^−1^·FW)) of the seashore paspalum was measured according to type of biochar, type of treatment, and irrigation level. To strengthen quality soil through organic amendment addition, we evaluated pH and EC (electrical conductivity (μS/cm)) at the beginning and end of planting using the method of Blakemore et al. [20].

The stress treatment lasted 10 months. At the end of the experiment, fresh biomass was weighed and then dried at 80 °C. Concerning the analytical techniques, two sophisticated methods were used in this study: factor analysis of mixed data (FAMD) and multiple factorial analysis (MFA). These two multivariate methods have the advantage of studying the overall variability and covariability of numerical variables (parameters) and categorical variables (experimental conditions).

### 2.4. Gas Exchange and Chlorophyll Pigment Content

Throughout the day, fresh leaves were inserted into the infrared gas analyzer (IRGA) (LCi-Portable Photosynthesis System, ADC, Hertfordshire, UK) between 9.00 a.m. and 12.00 a.m. to measure physiological variables. Each treatment received three readings [13,21].

In three biological duplicates, young leaves of *P. vaginatum* seedlings were evaluated to retrieve chlorophyll measurements following the protocol described in [13,20].

### 2.5. Relative Water Content (RWC)

The method described in [13,21,22,23] demonstrated how to determine the relative water content (RWC) of *P. vaginatum* leaves.

### 2.6. Leaf Stomatal Traits

Stomata impressions of *P. vaginatum* leaves were made using the technique described in [13].

### 2.7. Anatomical Characters

Using the technique described in [21], the anatomical characteristics of *P. vaginatum* were measured during winter, spring, and summer 2019/2020.

### 2.8. Chemometric Analyses

In this study, several types of parameters (physiological, stomatal, and anatomical) were measured according to the variation in the experimental conditions such as type of treatment, type of biochar, type of soil, and irrigation. Statistical analysis for relevant information extraction was necessary. Thus, the use of ANOVA as a statistical method to examine the effects of change in these conditions on the measurement of all the parameters is not judicious. Indeed, beyond the two-factor ANOVA, the statistical study loses its effectiveness; moreover, the study of each parameter according to the variation in two conditions requires considerable time and much space in the document. For these reasons, we prefer to use general multivariate methods, which allow us to study simultaneously the variability and covariability of all parameters according to the modified conditions of the experiments.

Two sophisticated analytical techniques were used in this study: factor analysis of mixed data (FAMD) and multiple factorial analysis (MFA). These two multivariate methods have the advantage of studying the overall variability and covariability of numerical variables (parameters) and categorical variables (experimental conditions).

FAMD is a principal component method dedicated to analyzing a data set containing both quantitative and qualitative variables; it allows us to explore the association between all variables, whether they are quantitative or qualitative [24].

MFA is a multivariate data analysis method for summarizing and visualizing a complex data table in which individuals are described by several sets of variables (quantitative and/or qualitative) structured into groups. It takes into account the contribution of all active groups of variables to define the distance between individuals [25].

Hence, in plastic pot tests, FAMD is utilized to compare the effects of three biochars on physiochemical properties and pigment. Furthermore, MFA is the best statistical approach for this case study because the data contain both mixed method variables organized into groups. MFA was used to observe how different agricultural waste biochars affected seasonal physiological changes in seashore paspalum under two different conditions: well-watered and drought.

The FactoMineR and Factoextra packages were used to perform all statistical calculations in RStudio Version 1.4.1717.

## 3. Results and Discussion

### 3.1. Effect of Three Biochars on Soil Properties

In aspects of the significant contribution of the various groups, pH and period played an important role in axis 1. This indicated that the main modes (pH and period) had a significant impact on all measurements, particularly the distribution of the physicochemical parameters analyzed.

The group “EC” (electrical conductivity), type biochar, and type treatment all made significant contributions to FAMD axis 2 (Figure 1a).

The effect of three biochars made from agricultural wastes on pH and EC of the sandy loam soil was explored in this research. The pH of soil amended with walnut shells biochar and peanut hull biochar was relatively higher than that of treatment options using almond shells biochar (Figure 1b and Figure 2). The peanut hull biochar and walnut shells biochar might be suggested to neutralize the acidic soils due to their richness in ash content. Additionally, almond shells biochar can benefit basic soils. In general, because our three biochars are all alkaline [3], they tend to neutralize the modified treatments. In line with our findings, Castaldi et al. [26] found that adding alkaline biochar amendment (3 and 6 kg m^−2^) to field soil planted with wheat led to pH increasing from 5.2 to 6.7 during the course of two growing seasons.

In addition, plastic-pot soil amended with walnut shells biochar and peanut hull biochar treatment revealed high values of EC, while the addition of almond shells biochar to soil did not show any increase after one year of biochar addition. In a UK field trial, Jones et al. [17] discovered that three years of biochar addition did not affect soil electrical conductivity.

### 3.2. Effect of Three Biochars on Pigment Content of P. vaginatum

Figure 3a demonstrates that chlorophyll a, chlorophyll b, total chlorophyll, and biochar type all made an enormous contribution to axis 1 but not axis 2. This tends to mean that these parameters seemed to have a highly significant effect on datasets located along axis 1, specifically on the dispersion of pigment group values measured in the final months of the different tests. 

In this context, carotenoids (car) and treatment type had a greater effect on FAMD axis 2. The two groups are noticeable on the FAMD’s axis 2 for distinguishing between the treatments studied (Figure 3a).

The correlation of the multiple factors—chla, chlb, total chlorophyll, carotenoids, and the FAMD factorial axes—is revealed by the loading plot shown in Figure 3b. The vector and loadings plot were used in this study to measure the effect of peanut hull biochar, almond shells biochar, walnut shells biochar, and addition of compost mixtures on pigment chlorophyll of *P. vaginatum* under two water conditions in the spring season (Figure 3b).

The exact locations of the four-factor groups were approximated using the first two important components of the FAMD (Dim 1 and 2 on the diagram), which represented 31.1% of the overall variability of the different components (Figure 3b). The depiction of the FAMD’s treatments is portrayed in Figure 3c. The therapies were completely distinguished by factorial axis 1 (29.4% of the variance), relying on chlorophylls (chla, chlb, chla + b) and carotenoid. Paspalum leaves cultivated in treatments amended with peanut hull biochar (3B0C_20 and 3B0C_60) and almond shells biochar (3B0C_60, 3B0C_20, 6B0C_20, 6B3C_60, and Test_20) had higher chlorophyll levels than walnut shells biochar. The second axis of FAMD revealed a high value of carotenoids for paspalum cultivated at the amended treatments (Figure 3c). Our findings confirmed that adding the three biochars to the modified treatments increased chlorophyll levels compared to controls, which agrees with the predictions of Fetjah et al. [27]. In line with our results, Xu et al. [28] found that peanut hull biochar increased chlorophyll content and photosynthesis.

Rehman et al. [29] reported that chlorophyll content played a large part in increasing biomass production, which explains why biochar application increased chlorophyll and biomass of *P. vaginatum*. With the walnut shells biochar application, leaves of paspalum did not show a high content of pigments. This outcome can be attributed to inadequate brightness captured by seashore paspalum during the spring season.

When subjected to severe drought, the pigment content of leaves of paspalum cultivated in certain amended treatments increased significantly when compared to control plant leaves. This discovery is consistent with the findings of several studies [27,28,30].

### 3.3. Effect of Three Biochars on Physiological Performance

During three seasons (winter, spring, and summer) using plastic pots, the effects of different percentages of three biochars on the physiological characteristics of *P. vaginatum* were studied (Figure 4a). Several characteristics were evaluated, such as relative water content, stomatal conductance, photosynthesis, fresh weight of aerial paspalum biomass (FWBA), chlorophyll parenchyma thickness (CPT), and total parenchyma thickness (TPT). MFA was performed to examine the whole effect of multiple agricultural waste biochars, trial season (winter, spring, and summer), and condition (well-watered—60% WHC and limited water—20% WHC) on physiological data recorded. 

In terminology about how each group of factors makes a major contribution to axis 1, distinct ramifications can be attained again for the performance of different sets of variables by the addition of activated carbon and compost. The contribution of relative water content appears to be the most significant for axis 1, followed by the contribution of weight. In Figure 4a, the much more notable contribution to axis 2 of the MFA would seem to have been from stomatal leaf traits, accompanied by photosynthesis and leaf anatomy (physiological). In this study, MFA was initially used to study the effect of each biochar on physiological characteristics, which are classified into five homogeneous groups, including photosynthesis, stomatal traits, fresh weight, relative water content, and leaf anatomy (CPC, TPT) (Figure 4b). The categorical group includes seasonal changes together with the proportion or rate of amendment (Figure 5). A plot of the groups was used by the five sets of changeable coordinates (group representation; Figure 5). The measurements were calculated using the MFA’s first two components (shown as Dim 1 and 2 in the figure), which compensated for 20.1 and 15.4% of the dataset’s total variance, respectively. 

The addition of the combination of biochar and compost in the sandy loam soil significantly increased paspalum growth. The four treatments (3B6C2Cws, 6C0B2Cpeh, 3B6C1_Calm, and 6C0B_1Calm) had a highly fresh weight of aerial biomass FWBA and relative water content at midday and predawn (individual score plot, Figure 5). Interestingly, the relative contribution of peanut hull biochar to improved paspalum growth was significantly higher than in the sandy loam soil substrate during the summer period, as indicated by an MFA study. The improvement in photosynthesis and the relative water content can be enhanced markedly by the addition of 6% peanut hull biochar to sandy loam soil. Peanut hull biochar addition led to enhanced plant growth as MOt and COt were higher in the biochar compared to the two other biochar. During the summer period, the seashore paspalum requires increased water to maintain the moisture level in the soil. In this study, the application of 6% peanut hull biochar at 20% WHC revealed the greatest result in economizing the water requirement. Indeed, the water-holding capacities (WHC) were 0.064, 0.122, 0.074, and 0.078 gH_2_O g^−1^ soil (dry weight) in 0% (control) and 6% peanut hull biochar, almond shells biochar, and walnut shells biochar), respectively. The biochar application increased the WHC by 47.54% in 6% peanut hull biochar, 13.51% in 6% almond shells biochar, and 17.94% in 6% walnut shells biochar compared to the control. Since peanut hull biochar is rich in carbon and organic matter, it had the best result in retaining water. Its application into the soil could have a beneficial result for growing healthy turfgrass in dry climates with minimal irrigation, which was characterized by 20% WHC in this study (Figure 4 and Figure 5).

The increase in A could be explained by increased leaf stomatal conductance and transpiration E following the different biochar amendments. The continuing improvement in gs and E might be linked to rising soil-water-holding capacity, which could also be credited to the highly permeable physical structure of the three biochars [3].

Actually, the three biochars added to the sandy loam soil did take time to disintegrate and improved all of the analyzed properties, especially during the summer season [6,28,31]. Xu et al. [28] showed that application of peanut hull biochar enhanced the growth of peanut, which agrees with our findings. Aside from that, test remedies containing NPK and soil would be used as a support material to compare the optimal mix of biochar and compost on paspalum growth regulation. Furthermore, the treatment modalities in the right side of Figure 5 were the best configurations of amendments used. The fresh weight biomass and photosynthesis had greater values with amended soils. The biochar was found throughout the summer season, confirming the gradual degradation of biochar [32]. Figure 4b and Figure 5 depict portions of each diagnosis that demonstrated anatomy value changes of seashore paspalum leaf exacerbated by environmental stresses. Underwater stress in the summer, the thickness of chlorophyll parenchyma, and total parenchyma increased significantly with different treatments.

Our chemometric analysis results, which used only ecophysiological characteristics throughout the year, obviously support the concept that biochars and compost certainly boost seashore paspalum species and strengthen drought resistance in semiarid and arid regions.

During the winter and spring seasons, all axes of MFA reveal that the control treatment (CTR) seemed to have low values of physiological traits and relative water content, but moderate values of stomatal traits.

In winter and spring seasons, seashore paspalum widely cultivated in amended conditions had lesser photosynthetic activity, reduced water availability (WUE, RWC), and elevated stomatal trait values. Furthermore, stomatal density (SD) increased with treatments amended with combined compost and biochar (3B3C_1Bpeh) or with compost alone (3C0B_1Bws). During the summer season, SD dramatically decreased in the modified treatments (Figure 5). Furthermore, *P. vaginatum* expansion was fairly low in the winter than in the warmer months, which is similar to the findings of Liu et al. [33] that suggest bamboo biochar increased rice agricultural output during two crop seasons.

As previously mentioned, leaf stomatal density had been increased in *Solanum melongena* under drought [34]. In this sense, Xu and Zhou [35] discovered that throughout moderate rainfall, stomatal density increased while total water declined, whereas, during dry seasons, SD lessened considerably, suggesting that grass did appear to adapt to its surroundings through leaf flexibility. Furthermore, Tanure et al. [36] proved that the incorporation of biochar diminished photosynthetic activity, stomatal conductance, and leaf RWC under dry season conditions. Abideen et al. [30] reported that adding 2.5 percent biochar reduced RWC, thereby mitigating relative water content for plants. Furthermore, Paneque et al. [37] discovered a decline in stomatal conductance in sunflower seedlings after biochar application in the soil and under water stress. Despite this, Abideen et al. [30] found that using biochar led to improved growth and photosynthesis during water stress.

## 4. Conclusions

Overall, our findings pointed out various responses of seashore paspalum to seasonal changes, depending on three biochars under two water conditions (60 percent and 20 percent WHC). The effect of deficit irrigation on physiological characteristics was stronger in unamended soils than in amended soils with biochar and/or compost. Soil amendments resulted in increased EC and pH of soil amended with peanut hull biochar compared to walnut shells biochar and almond shells biochar. The addition of 6% of the peanut hull biochar at 20% of WHC revealed a high response of soil fertility to maintain nutrients to paspalum even in harsh conditions. 

In addition, during the summer season, the species tended to increase physiological characteristics under water stress, mainly through the use of peanut hull biochar to soil compared to the two other biochars. This study showed the greatest implication for turf managers seeking to expand seashore paspalum into dry climate areas. Under these conditions, high-value turfgrass management options are frequently limited and exorbitant. 

As the world’s water scarcity worsens due to climate change, increasing the use of biochar and compost mixtures for turf landscape irrigation will be necessary to reduce water requirements. We believe that our research confirms that choosing a biochar type will be one of the solutions to water shortages in arid zones. The additional information provided here enriches our understanding of real-world conditions. It was also suggested that biochar made from peanut hull should be tested in situ to determine its efficacy for increasing irrigation efficiency on golf courses.

## Figures and Tables

**Figure 1 biology-11-00560-f001:**
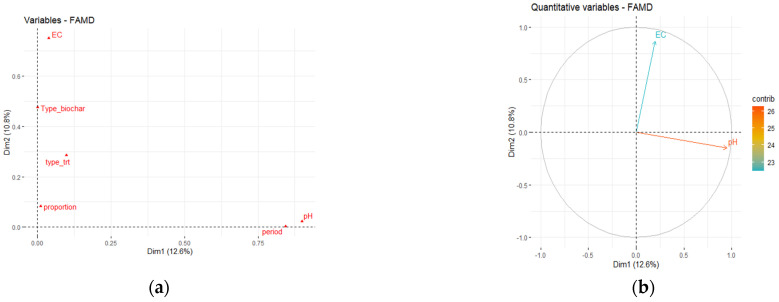
(**a**) Factor analysis of mixed data (FAMD) was used to analyze physicochemical data from *Paspalum vaginatum* Swartz. Electrical conductivity, pH, type of treatment, period, and proportion are represented as variable groupings. (**b**) Loading plot of *P. vaginatum* pH and electrical conductivity during three seasons and amended with three biochars.

**Figure 2 biology-11-00560-f002:**
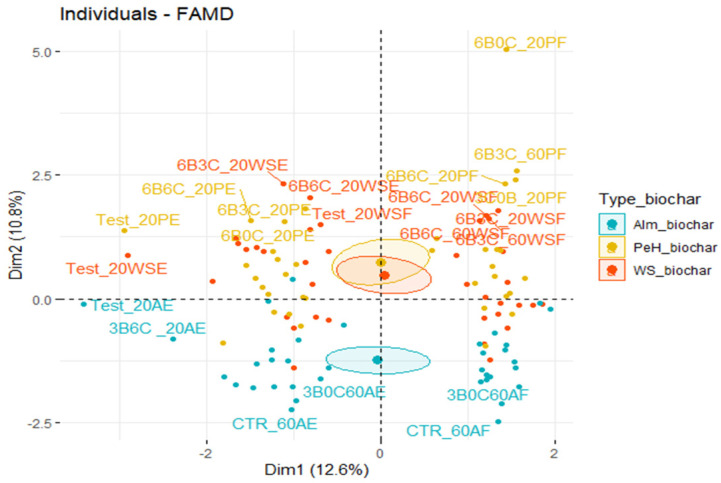
Score plot demonstrating the impact of multiple biochars on soil pH and electrical conductivity in a plastic pot at the start of planting and the conclusion of the research.

**Figure 3 biology-11-00560-f003:**
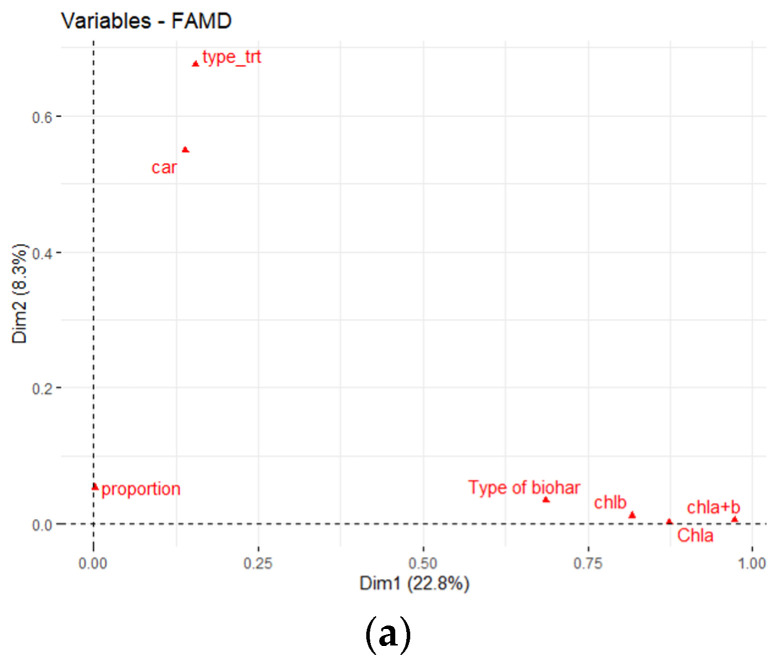
Physicochemical parameters of *P. vaginatum* FAMD during the spring season. (**a**) Under two different water conditions, four groups of *P. vaginatum* different factors were characterized. (**b**) The variable loading plot of the two first principal components. (**c**) The score plot describes the effect on the two first principal components for the plastic-pot treatment options when using almond shell biochar, walnut shell biochar, and peanut hull biochar.

**Figure 4 biology-11-00560-f004:**
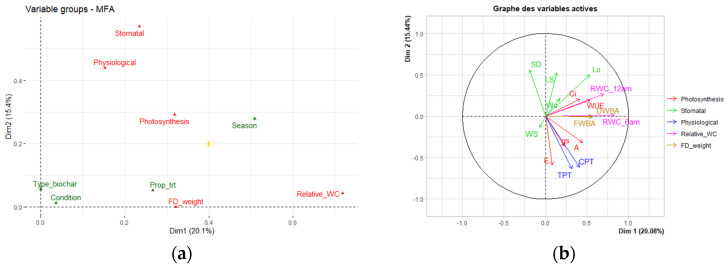
MFA of *P. vaginatum* physiological and stomatal parameters during three seasons. (**a**) Under two different water conditions, five groups of different factors were characterized for *P. vaginatum* under two water shortages (20 and 60% of WHC). (**b**) Variable loading plot of the two first principal components.

**Figure 5 biology-11-00560-f005:**
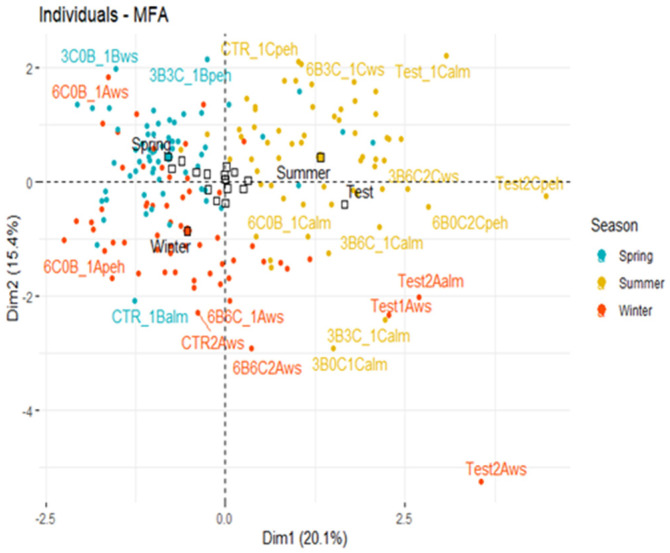
MFA of *P. vaginatum* physiological measurements during three seasons. Key: A—winter season, B—spring season, C—summer season, 1—60% WHC, 2—20% WHC; ws—walnut shells biochar, peh—peanut hull biochar, alm—almond shells biochar.

## Data Availability

The data in this study are available upon reasonable request to the corresponding author.

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
