# Peer review of "Seasonal Paspalum vaginatum Physiological Characteristics Change with Agricultural Byproduct Biochar in Sandy Potting Soil"

_biology, 2022, doi:10.3390/biology11040560_

Round 1
Reviewer 1 Report
This study demonstrates the impact of three biochars on enhancing physiological and stomatal traits characteristics of seashore paspalum under two water shortages during the summer season compared to other seasons. The authors used a good number of traits to assess water shortages in seashore paspalum. So, I appreciate the authors efforts in analyzing a wide set of traits.
I have found some weaknesses in the manuscript:
- ANOVA analysis should be calculated for between treatments to assess the interactions.
- Multicollinearity analysis should be calculated between traits.
- Should be measure Kaiser-Meyer-Olkin (KMO) of sampling adequacy
- Should show the table of factor analysis to know the effective traits.
Author Response
Responses to Reviewer Comments
Reviewer 1
First, we want to thank Reviewer #1 for his/her positive reaction to the overall content of the review. We are also grateful for the remarks and constructive suggestions. Below is a point-by-point description of how we revised the manuscript.
This study demonstrates the impact of three biochars on enhancing physiological and stomatal traits characteristics of seashore paspalum under two water shortages during the summer season compared to other seasons. The authors used a good number of traits to assess water shortages in seashore paspalum. So, I appreciate the authors efforts in analyzing a wide set of traits.
I have found some weaknesses in the manuscript:
- ANOVA analysis should be calculated for between treatments to assess the interactions.
The two-factor analysis of variance is performed on each parameter measured under conditions of changing treatment type. The treatments used in this study are:
1- "Amendment mixtures" which take 8 modalities (8 preparations) plus a Control and a test.
This factor was given the name "type_trt" in the article
2- "Biochar" which takes three modalities: peanut hull, walnut shells and almond shells.
This variable has the name "type_biochar".
The two-factor analysis of variance reveals the significance (at the 0.05 risk level) of the effects of changing the modality of each variable and the effect of their interaction on the measurement of physiological, photosynthesis, stomatal, and pigment parameters. The results are summarized in the table below:
A clear very significant effect of changing the modality of the "type_biochar" condition on the majority of the measured parameters is detected.
A significant effect on some parameters of the "treatment type" condition.
The effect of the interaction between the two treatment factors is not significant.
It should be noted that these results confirm the conclusions obtained by the FAMD and MFA methods used. Indeed, according to the study of the effects of the conditions on the parameters of Pigment by FAMD (figure 5-a), according to axis 1 a perfect correlation is distinguished between the three parameters Chla, Chlb and Chlab and the factor type_biochar. On the other hand, according to axis 2, a good correlation is established between the parameter car and the factor type_trt (even the ANOVA significance test does not indicate this).
Regarding the MFA study, it was found that the photosynthesis, stomatal and physiological parameters, as well as the type_trt factor are well correlated with the MFA axis 2. The same can be said for the parameters of weight biomass and RWC and the factor type_biochar, which are well correlated with axis 1 of MFA.
|
|
Parameters |
p-value |
||
|
Type-trt |
Type-biochar |
Interaction |
||
|
Photosynthesis |
E |
0,011 |
<0,000 |
0,615 |
|
A |
0,036 |
0,057 |
0,660 |
|
|
Gs |
0,023 |
0,002 |
0,629 |
|
|
WUE |
0,986 |
0,001 |
0,762 |
|
|
Ci |
0,477 |
0,069 |
0,648 |
|
|
Stomatal |
SD |
0,056 |
<0,000 |
0,522 |
|
LS |
0,908 |
0,300 |
0,094 |
|
|
WS |
0,115 |
0,007 |
0,004 |
|
|
Physiological parameters |
CPT |
0,418 |
<0,000 |
0,639 |
|
TPT |
0,284 |
<0,000 |
0,107 |
|
|
RWC |
RWC_6am |
0,029 |
0,289 |
0,971 |
|
RWC_12am |
0,226 |
0,011 |
0,436 |
|
|
Weight of biomass |
FWBA |
<0,000 |
0,7149 |
0,731 |
|
DWBA |
<0,001 |
0,7345 |
0,7377 |
|
|
Pigment |
Chla |
0,025 |
<0,000 |
0,343 |
|
Chlb |
0,794 |
<0,000 |
0,165 |
|
|
Chla+b |
0,038 |
<0,000 |
0,082 |
|
|
|
|
|
|
|
- Multicollinearity analysis should be calculated between traits.
The following collinearity matrix, calculated for the pigment parameters measured under the different conditions, shows a perfect correlation between the Chla, Chlb and Chla+b measurements:
|
|
Chla |
chlb |
chlab |
car |
|
Chla |
1 |
|||
|
Chlb |
0,735 |
1 |
||
|
Chlab |
0,962 |
0,892 |
1 |
|
|
Car |
0,265 |
0,262 |
0,282 |
1 |
With the help of the psych package under R, we obtained the correlation matrix, of all the characteristic parameters of the study, indicating that there are indeed multicolinearities between them:
- Should be measure Kaiser-Meyer-Olkin (KMO) of sampling adequacy
The analysis of our data was done by a Multiple Factorial Analysis (Weighted PCA) where the variables (measured parameters) are organized in groups. The adequacy on the consistency of the data was provided from two statistical methods namely the Kaiser-Meyer-Olkim (KMO) index and Bartlett's test of sphericity :
The Kaiser-Meyer-Olkin (KMO) adequacy index
The Measure of Sampling Adequacy (MSA) or Kaiser-Meyer-Olkin (KMO) indicates the extent to which the items selected form a coherent whole and adequately measure a concept. This index varies between 0 and 1. According to Stafford and Bodson (2006), a KMO value below 0.50 is unacceptable. Thus, a KMO value must be greater than 0.50 for factor analysis to be feasible. The measurement of the KMO index was done with the statistical software RStudio by the function KMO() of the package EFA tools which gave a global index of KMO equal to 0.631.
Bartlett's test of sphericity
Bartlett's sphericity test verifies the null hypothesis that all correlations are equal to zero, i.e. that the variables are not correlated with each other and, therefore, are perfectly independent of each other.
The chi-square value calculated for a ddl equal to 120, by the cortest.bartlett( ) function of the psych package, is found to be equal to 1988.166 and a p-value that tends towards 0, which allows us to reject the null hypothesis and to judge the significance of the correlations between the variables of our data.
Thus the conditions of the factorial analysis are fulfilled, and thus the analysis of our data by MFA is feasible and acceptable.
- Should show the table of factor analysis to know the effective traits.
Table of group contribution
|
Groups |
Dim.1 |
Dim.2 |
||||
|
Coord |
contrib |
cos2 |
Coord |
contrib |
cos2 |
|
|
Photosynthesis |
0.32 |
18.25 |
0.06 |
0.29 |
21.65 |
0.05 |
|
Stomatal |
0.23 |
13.45 |
0.02 |
0.57 |
42.42 |
0.14 |
|
Physiological |
0.15 |
8.74 |
0.02 |
0.44 |
32.72 |
0.19 |
|
Relative_WC |
0.72 |
41.16 |
0.49 |
0.04 |
3.21 |
0.00 |
|
FD_weight |
0.32 |
18.40 |
0.10 |
0.00 |
0.00 |
0.00 |
Table of variable supplementary
|
Supplementary |
Dim.1 |
Dim.2 |
||
|
Coord |
cos2 |
Coord |
cos2 |
|
|
Prop_trt |
0.27 |
0.01 |
0.05 |
0.00 |
|
Type_biochar |
0.00 |
0.00 |
0.06 |
0.00 |
|
Season |
0.51 |
0.13 |
0.28 |
0.04 |
|
Condition |
0.04 |
0.00 |
0.01 |
0.00 |
|
variables numériques |
Dim.1 |
Dim.2 |
||||
|
Coord |
contrib |
cos2 |
Coord |
contrib |
cos2 |
|
|
E |
0.08 |
0.17 |
0.01 |
-0.59 |
11.44 |
0.35 |
|
A |
0.45 |
5.02 |
0.20 |
-0.32 |
3.28 |
0.10 |
|
gs |
0.23 |
1.34 |
0.05 |
-0.36 |
4.24 |
0.13 |
|
WUE |
0.54 |
7.37 |
0.29 |
0.19 |
1.25 |
0.04 |
|
Ci |
0.42 |
4.34 |
0.17 |
0.21 |
1.44 |
0.04 |
|
SD |
-0.19 |
1.33 |
0.04 |
0.56 |
14.72 |
0.31 |
|
LS |
0.14 |
0.67 |
0.02 |
0.53 |
12.94 |
0.28 |
|
WS |
-0.07 |
0.18 |
0.01 |
-0.13 |
0.85 |
0.02 |
|
Lo |
0.53 |
10.22 |
0.28 |
0.50 |
11.79 |
0.25 |
|
Wo |
0.17 |
1.05 |
0.03 |
0.21 |
2.11 |
0.04 |
|
CPT |
0.41 |
5.39 |
0.17 |
-0.61 |
15.86 |
0.38 |
|
TPT |
0.32 |
3.35 |
0.10 |
-0.63 |
16.86 |
0.40 |
|
RWC_6am |
0.83 |
23.92 |
0.69 |
0.01 |
0.01 |
0.00 |
|
RWC_12am |
0.70 |
17.24 |
0.49 |
0.27 |
3.21 |
0.07 |
|
FWBA |
0.57 |
9.18 |
0.32 |
-0.01 |
0.00 |
0.00 |
|
DWBA |
0.57 |
9.22 |
0.32 |
0.00 |
0.00 |
0.00 |

Reviewer 2 Report
The research article entitled, “Improving seasonal physiological characteristics and stomatal traits in seashore paspalum through the use of three biochars derived from agricultural wastes" evaluated the influence of biochar derived from peanut hull, walnut shells, and walnut shells on physio-logical, stomatal traits and growth of paspalum. Authors determined paspalum leaves gas exchange characteristics, stomatal traits, anatomy traits, relative water content, and paspalum pigments. They found that MFA factor map spotlighted the net effect of the experiment in pot and seasonal changes obtained through the experience of the physiological traits of paspalum vaginatum, as well as pH and EC of soil data.
Altogether this is an important and timely research article, this reviewer has certain suggestions that would help produce a more comprehensive overview of the topic:
Comments:
1, The English of manuscript can be polished (minor) and typo errors can be checked.
2, Authors can add one paragraph for abbreviations.
3, Briefly discuss the translational efficacy of their study and future direction to this area of research.
4, Authors can include the limitations to their study.
5, At least one illustrative figure may be provided as to highlight the summary of this study.
6, Figure 3A quality may be improved (high resolution).
Author Response
Reviewer 2
We would like to express our gratitude to reviewer #2 for his/her insightful comments and thorough review, which were extremely helpful in revising the manuscript.
The research article entitled, “Improving seasonal physiological characteristics and stomatal traits in seashore paspalum through the use of three biochars derived from agricultural wastes" evaluated the influence of biochar derived from peanut hull, walnut shells, and walnut shells on physio-logical, stomatal traits and growth of paspalum. Authors determined paspalum leaves gas exchange characteristics, stomatal traits, anatomy traits, relative water content, and paspalum pigments. They found that MFA factor map spotlighted the net effect of the experiment in pot and seasonal changes obtained through the experience of the physiological traits of paspalum vaginatum, as well as pH and EC of soil data.
Altogether this is an important and timely research article, this reviewer has certain suggestions that would help produce a more comprehensive overview of the topic:
Comments:
1, The English of manuscript can be polished (minor) and typo errors can be checked.
· As suggested by the reviewer, we improved the whole manuscript and we corrected some spelling and grammar mistakes in order to get the attention it deserves
2, Authors can add one paragraph for abbreviations.
· As suggested by the reviewer, we added all abbreviations in Material and Methods.
3, Briefly discuss the translational efficacy of their study and future direction to this area of research.
· As suggested by the reviewer, we added the perspectives of our study in the conclusion.
5, At least one illustrative figure may be provided as to highlight the summary of this study.
· As suggested by the reviewer, we provided a graphical abstract to summarize the study.
6, Figure 3A quality may be improved (high resolution).
- As suggested by the reviewer, we improved the quality of the figure.

Reviewer 3 Report
General:
English composition: the text meaning is understandable but could be much smoother and more efficient (less verbose) if proofed by a professional science editor. Improper and over-use of conjugations to start paragraphs and sentences, single-sentence paragraphs, overly long paragraphs with multiple topics, and punctuation are prime examples.
The authors collected and analyzed quite a bit of data that merits publication. However, the M&M are incomplete and the R&D as well as Conclusions could have 80% greater interpretive value for future readers. The entire manuscript needs to be re-organized and the focus taken off biochar production and back to its original purpose: Answer the question of whether and why we should use which biochar or not to grow healthy turf grass in dry climates with minimal irrigation. It seems like the environmental engineer dominated the writing and the plant and soil scientists did not participate much.
Title: The title is long, incomplete and misleading. Suggest something like: “Seasonal Paspalum vaginatum physiological characteristics change with agricultural byproduct biochar in sandy potting soil”
Simple summary:
line 26 summarize the practical results of this research: which biochar, if any, improved grass tolerance to soil-moisture stress physiology, in what season and to what an extent? How much irrigation could conceivably be saved on turfgrass in hot dry climates?
Abstract: Is woefully inadequate and seems to have been written at the last minute with little attention to detail. It is actually the most important part of the manuscript and, if published, will likely be the only part of the article most readers will look at. As such, I strongly suggest putting a little more effort into it.
- 28 “The influence of biochar” does not indicate that it was added to the soil. Suggest “The influence of soil-added biochar...”
- 29 “pot experiments during three seasons” is misleading. Was this a greenhouse or open-air trial? State as such. Seasonal differences are usually not an issue in greenhouse studies with moisture and temperature controls. If in the greenhouse, specify the temperature ranges for each season.
- 29 Replace “paspalum” with Paspalum vaginatum
- 33 Based on the conditions “at the two locations, we collected soil.” What two locations, why two locations? No previous mention. What “conditions”?
- 33 “the effect of carbon sequestration on soil properties was assessed” No previous mention of “soil carbon sequestration” was made. Do you mean added C from biochar? If so, don’t confuse readers—just call it C addition.
- 36 include specific results, including statistical significance or lack thereof.
- 37 End the abstract with practical implications. Add or don’t add biochar to benefit grass physiology in what season? Why should we be excited about these results and want to apply them in Cape Town, Buenos Aires, or Roma? If the authors don’t show an interest in the results, why should Biology publish them?
Introduction:
51 Why would “reducing the agricultural residues have a negative impact on soil carbon stores”? Seems the authors are confusing two issues here: 1. dealing positively with agricultural byproduct issues and 2. Increasing soil-C sequestration, thereby benefiting the environment. Mean what you write and write what you mean.
71 “when compared to walnut shells biochar and almond shells biochar, the maximum benefit would have been expected to be obtained with peanut hull biochar, which could potentially provide biological properties.” Why do you hypothesize that peanut biochar would be better? Make a clearer argument for this, based on your literature review.
74 Replace “The effects of three biochars on the physicochemical properties of soils and the physiological responses of seashore paspalum are discussed in this article.” with 2 to 3 clear objective statements that your specific research addressed.
M&M
There are numerous details missing that make it impossible to duplicate the trial. These must be added and organized cohesively. Example: was this a greenhouse trial? Or open-air table trial? Shade cloth? Solar radiation? Average/high/low season temperatures? Relative humidity? What sort of water was used to irrigate? Chemical analyses (to determine if nutrient were added or pH changed by irrigation). How large were the pots and how much soil (dry matter basis) was placed in each pot? What was the soil chemical analyses (Mehlich III for plant-available and total for at least N and P)? Etc. etc.
Biochar biochemical and physiochemical properties arguable are not the result of your treatments. Rather, they are descriptions of your materials (biochar). The first section of R&D should therefore be moved to this section.
As organized, your M&M are impossible to follow or understand. Reorganize in a logical fashion.
- Experimental design
- Independent variables
- Dependent variables
- Statistical analyses
Line 77: start your M&M section with a clear experimental design. How many factors? How many treatments within each factor? How were these arranged in the greenhouse? RCBD? How many replications? What was your experimental unit (the unit that gave you each individual dependent variable).
Line 77: then list and describe your independent variables (your treatments, organized by factors and each of their treatments).
Line 77: then list your dependent variables (what you actually measured)
Line 83 “same soil” same to what?
Line 82: “2.2. Experimental design” this section is NOT a description of the experimental design. It is the actual M&M. See above for what is needed to describe the experimental design.
Line 93, 98 etc.: Verbs in M&M should be conjugated in past tense.
Line 89: 198,256,343 °C this temperature is not physically possible. Did you mean to separate them with spaces so that these are actually 3 temperatures, a different one for each agricultural byproduct? If so, explain why the difference among them.
Line 136: multiple factorial analysis (MFA) is a good example of why a clear experimental design paragraph at the start of M&M will avoid confusion. Which factors are the authors talking about here? They were never identified as such. Also, clearly write out a stats model.
R&D
Line 145 to 198 “3.1. Characterization of the three biochars” This entire section is not a result of a treatment. It is a description of your materials (biochars).
Line 310 avoid subjective terms such as “vast” and stick to facts. What percentage greater of which nutrients were available during the summer compared to the two other seasons? Be specific.
Conclusions
Line 353 “In summary,” Conclusions are NOT a summary (repeat) of your M&M and R&D. They should, rather, provide explicit details of what you found that expanded our scientific envelope throughout the scientific community and why they are important to the international community that reads this journal. Get excited about your results!! If you don’t, how do you expect future readers to care what you identified as the important contributions?
357 “This result ensures our hypothesis in this survey.” First, this is not a survey. Second, results don’t ensure a hypothesis. They either lead you to accept or reject the hypothesis or hypotheses.
35 “In this study, MFA factor map spotlighted the net effect of the experiment in pot and seasonal changes obtained through the experience of the physiological traits of paspalum vaginatum, as well as pH and EC of soil data.” You are enamored with your results but present no conclusions. MFA factor map spotlighted WHAT net effects that tell us WHY we should or should not use biochar on turfgrass in dry climates?
Author Response
Reviewer 3
We sincerely thank reviewer #3 for his/ her valuable comments and thorough review, which were of great help in revising the manuscript. Accordingly, the revised manuscript has been systematically improved with new information and additional interpretations. The answers to his/her specific comments/suggestions are as follows.
English composition: the text meaning is understandable but could be much smoother and more efficient (less verbose) if proofed by a professional science editor. Improper and over-use of conjugations to start paragraphs and sentences, single-sentence paragraphs, overly long paragraphs with multiple topics, and punctuation are prime examples.
· As suggested by the reviewer, we improved the whole manuscript and we corrected some spelling and grammar mistakes with an English professor in order to get the attention it deserves.
The authors collected and analyzed quite a bit of data that merits publication. However, the M&M are incomplete and the R&D, as well as Conclusions, could have 80% greater interpretive value for future readers. The entire manuscript needs to be re-organized and the focus taken off biochar production and back to its original purpose: Answer the question of whether and why we should use which biochar or not to grow healthy turf grass in dry climates with minimal irrigation. It seems like the environmental engineer dominated the writing and the plant and soil scientists did not participate much.
· As suggested by the reviewer, we improved the whole manuscript and we completed M&M are incomplete and the R&D, as well as Conclusions. Also, we answer the question why we should use which biochar or not to grow healthy turf grass in dry climates with minimal irrigation in the R&D part.
Title: The title is long, incomplete and misleading. Suggest something like: “Seasonal Paspalum vaginatum physiological characteristics change with agricultural byproduct biochar in sandy potting soil”
· As suggested by the reviewer, we changed the title.
Simple summary:
line 26 summarize the practical results of this research: which biochar, if any, improved grass tolerance to soil-moisture stress physiology, in what season and to what an extent? How much irrigation could conceivably be saved on turfgrass in hot dry climates?
· As suggested by the reviewer, we calculated the water holding capacity of the soil amended with 6% of biochar , which was the highest amount of peanut hull biochar.We found that the application of 6% of peanut hull biochar had increased the water holding capacity of amende soil compared to control.Also the analysis of peanut hull biochar revealed its richness by organic carbon with53.53% and a high total organic matter by 92.29%.All these virtues ,has made the addition of 6% peanut hull biochar to soil improved grass tolerance to soil-moisture stress physiology in the summer season. In fact, due to its ability to retain water, the application of peanut hull biochar to soil could potentially save 47,54 percent on turfgrass in hot, dry climates.
Abstract: Is woefully inadequate and seems to have been written at the last minute with little attention to detail. It is actually the most important part of the manuscript and, if published, will likely be the only part of the article most readers will look at. As such, I strongly suggest putting a little more effort into it.
- 28 “The influence of biochar” does not indicate that it was added to the soil. Suggest “The influence of soil-added biochar...”
- 29 “pot experiments during three seasons” is misleading. Was this a greenhouse or open-air trial? State as such. Seasonal differences are usually not an issue in greenhouse studies with moisture and temperature controls. If in the greenhouse, specify the temperature ranges for each season.
- 29 Replace “paspalum” with Paspalum vaginatum
- 33 Based on the conditions “at the two locations, we collected soil.” What two locations, why two locations? No previous mention. What “conditions”?
- 33 “the effect of carbon sequestration on soil properties was assessed” No previous mention of “soil carbon sequestration” was made. Do you mean added C from biochar? If so, don’t confuse readers—just call it C addition.
- 36 include specific results, including statistical significance or lack thereof.
- 37 End the abstract with practical implications. Add or don’t add biochar to benefit grass physiology in what season? Why should we be excited about these results and want to apply them in Cape Town, Buenos Aires, or Roma? If the authors don’t show any interest in the results, why should Biology publish them?
- As recommended by the reviewer, we improved the quality of the abstract. We edited several sentences and rectified some spelling and grammar issues, as indicated by the reviewer. Some sentences have been eliminated from the revised manuscript to give them the attention they deserve.
Introduction:
51 Why would “reducing the agricultural residues have a negative impact on soil carbon stores”? Seems the authors are confusing two issues here: 1. dealing positively with agricultural byproduct issues and 2. Increasing soil-C sequestration, thereby benefiting the environment. Mean what you write and write what you mean.
71 “when compared to walnut shells biochar and almond shells biochar, the maximum benefit would have been expected to be obtained with peanut hull biochar, which could potentially provide biological properties.” Why do you hypothesize that peanut biochar would be better? Make a clearer argument for this, based on your literature review.
74 Replace “The effects of three biochars on the physicochemical properties of soils and the physiological responses of seashore paspalum are discussed in this article.” with 2 to 3 clear objective statements that your specific research addressed.
- We improved the quality of the introduction, as suggested by the reviewer. As the reviewer suggested, we edited several sentences and corrected some spelling and grammar errors. Some sentences have been removed from the revised manuscript to give them the attention they deserve.
M&M
There are numerous details missing that make it impossible to duplicate the trial. These must be added and organized cohesively. Example: was this a greenhouse trial? Or open-air table trial? Shade cloth? Solar radiation? Average/high/low season temperatures? Relative humidity? What sort of water was used to irrigate? Chemical analyses (to determine if nutrient were added or pH changed by irrigation). How large were the pots and how much soil (dry matter basis) was placed in each pot? What was the soil chemical analyses (Mehlich III for plant-available and total for at least N and P)? Etc. etc.
Biochar biochemical and physiochemical properties arguable are not the result of your treatments. Rather, they are descriptions of your materials (biochar). The first section of R&D should therefore be moved to this section.
As organized, your M&M is impossible to follow or understand. Reorganize in a logical fashion.
- Experimental design
- Independent variables
- Dependent variables
- Statistical analyses
Line 77: start your M&M section with a clear experimental design. How many factors? How many treatments are within each factor? How were these arranged in the greenhouse? RCBD? How many replications? What was your experimental unit (the unit that gave you each individual dependent variable)?
Line 77: then list and describe your independent variables (your treatments, organized by factors and each of their treatments).
Line 77: then list your dependent variables (what you actually measured)
Line 83 “same soil” same to what?
Line 82: “2.2. Experimental design” this section is NOT a description of the experimental design. It is the actual M&M. See above for what is needed to describe the experimental design.
Line 93, 98 etc.: Verbs in M&M should be conjugated in past tense.
Line 89: 198,256,343 °C this temperature is not physically possible. Did you mean to separate them with spaces so that these are actually 3 temperatures, a different one for each agricultural byproduct? If so, explain why the difference among them.
- Done
Line 136: multiple factorial analysis (MFA) is a good example of why a clear experimental design paragraph at the start of M&M will avoid confusion. Which factors are the authors talking about here? They were never identified as such. Also, clearly write out a stats model.
R&D
Line 145 to 198 “3.1. Characterization of the three biochars” This entire section is not a result of a treatment. It is a description of your materials (biochars).
Line 310 avoid subjective terms such as “vast” and stick to facts. What percentage greater of which nutrients were available during the summer compared to the two other seasons? Be specific.
- Done
Conclusions
Line 353 “In summary,” Conclusions are NOT a summary (repeat) of your M&M and R&D. They should, rather, provide explicit details of what you found that expanded our scientific envelope throughout the scientific community and why they are important to the international community that reads this journal. Get excited about your results!! If you don’t, how do you expect future readers to care what you identified as the important contributions?
357 “This result ensures our hypothesis in this survey.” First, this is not a survey. Second, results don’t ensure a hypothesis. They either lead you to accept or reject the hypothesis or hypotheses.
35 “In this study, MFA factor map spotlighted the net effect of the experiment in pot and seasonal changes obtained through the experience of the physiological traits of paspalum vaginatum, as well as pH and EC of soil data.” You are enamored with your results but present no conclusions. MFA factor map spotlighted WHAT net effects that tell us WHY we should or should not use biochar on turfgrass in dry climates?
- As recommended by the reviewer, we improved the quality of the conclusion. we edited several sentences. Some sentences have been removed from the revised manuscript to give them the attention they deserve.
Round 2
Reviewer 1 Report
I agree with the reply of the authors and they provided satisfactory answers.
Author Response
We thank reviewer 1 very much.
Reviewer 3 Report
General:
The scientific merit is evident.
The manuscript contains fewer English errors but could still benefit from a professional editor. I simply don’t have the time to rewrite the entire manuscript.
Title:
Simple summary:
16 italicize Paspalum vaginatum
28 47,54 percent should be “...47.5% on irrigation amounts...”
Abstract:
31 ...Paspalum vaginatum (seashore paspalum) is critical...”
42 “...the ability of peanut hull biochar to facilitate seashore paspalum germination and establishment...”
43 47.5%
43 6B6C abbreviation has not been previously defined in abstract.
Keywords: add “seashore paspalum”
Introduction:
Split the introduction into three distinct paragraphs. Suggest starting new paragraphs at line 66 and 79.
79 This is not a survey. This is a research experiment.
M&M
The authors failed once again to start the M&M section with a clear description of their experimental design. How many factors were examined? I count at least 1. Seasons and 2. Biochar types 3. Biochar amount and 4. Irrigation level. How were the pots arranged on tables? Was each pot considered an experimental unit? How many replications?
R&D
Conclusions
484 delete “tremendous” since this is subjective and has not place in scientific writing.
485 start a new paragraph since this is a new topic.
492 start a new paragraph since this is a new topic.
493 delete “wisely” since this is subjective and has no place in scientific writing.
494 delete “nonetheless” since meaningless.
496 replace “”...will be tested to better represent local golf course conditions rather than limiting the technology to a specific potting trial.” with “...should be tested in situ to determine its efficacy for increasing irrigation efficiency on golf courses.”
